# Risk Factors of Long-Term Care Insurance Certification in Japan: A Scoping Review

**DOI:** 10.3390/ijerph19042162

**Published:** 2022-02-14

**Authors:** Shuko Takahashi, Yuki Yonekura, Nobuyuki Takanashi, Kozo Tanno

**Affiliations:** 1Department of Health and Welfare, Iwate Prefectural Government, Morioka 020-8570, Iwate, Japan; shutakahashi-iwt@umin.ac.jp; 2Division of Medical Education, Iwate Medical University, Yahaba-cho 028-3694, Iwate, Japan; 3Department of Critical Care, Disaster and General Medicine, Division of General Medicine, Iwate Medical University, Morioka 020-8505, Iwate, Japan; 4Department of Nursing Informatics, Graduate School of Nursing Science, St. Luke’s International University, Tokyo 104-0044, Japan; yyonekura@slcn.ac.jp; 5Department of Hygiene and Preventive Medicine, School of Medicine, Iwate Medical University, Yahaba-cho 028-3694, Iwate, Japan; pears_takanashi@ybb.ne.jp

**Keywords:** dementia, functional disability, Japan, long-term care, longitudinal studies

## Abstract

This study aimed to review evidence on future long-term care associated with pre-existing factors among community-dwelling Japanese older adults. We systematically searched cohort and nested case–control studies published between 2000 and 2019 that assessed long-term care certification using the PubMed, CINAHL, and EMBASE databases. The relationship between long-term care insurance information and risk factors was investigated. The protocol was registered with the Open Science Framework. We extracted 91 studies for synthesis, including 84 prospective cohort studies, 1 retrospective cohort study, and 6 nested case–control studies. Certification for long-term care was classified into two endpoints: onset of functional disability and dementia. There were 72 studies that used long-term care certification as a proxy for functional disability, and 22 used long-term care information to indicate the onset of dementia. Common risk factors related to functional disability were physical function, frailty, and oral condition. Motor function and nutritional status were common risk factors for dementia. We found consistent associations between premorbid risk factors and functional disability and dementia. The accumulation of evidence on the incidence of long-term care and associated factors can aid the development of preventive measures. Future studies should aim to integrate this evidence.

## 1. Introduction

As the number of older adults increases worldwide, the number of functional disabilities experienced owing to comorbidities may also increase, along with dementia. The Organization for Economic Cooperation and Development member countries reported that 10 million people were affected by dementia in 2000, of which 7% were aged 65 or older [1]. The number of individuals with disabilities in the United States is estimated to reach approximately 21 million by 2040 [2]. In Japan, the rising number of older adults is projected to increase rates of functional disability [3,4] and dementia [5], raising serious concerns regarding the increasing care burden. The chronic care model is widely adopted worldwide to improve chronic patient outcomes by changing patients’ care and managing chronic diseases [6,7]. Meanwhile, the government of Japan implemented the long-term care (LTC) insurance system in 2000, based on several European programs, to help older people with functional disabilities adequately manage their daily lives [8]. In 2017, about 6.4 million people received these certifications—a 2.5-fold increase over 17 years [9]—and LTC insurance expenditure reached JPY 9.4 trillion, from about JPY 4 trillion in 2000 [10]. Controlling the increase in the number of new LTC insurance certifications is an urgent health and financial challenge. It is necessary to determine the characteristics of people at risk of functional disability to establish effective preventive measures for functional disability.

To receive LTC certification, an older person first contacts the municipal government to apply for care needs. Trained local government officials then assess the individual’s degree of functional disability using a questionnaire developed by the Ministry of Health, Labour and Welfare. Then, officials calculate the individual’s standardized scores for physical and cognitive functions, based on the questionnaire, to determine whether the applicant is eligible for LTC insurance benefits. The Nursing Care Needs Certification Board decides whether an LTC insurance certificate should be issued, after considering the results of the initial assessment and using the primary care physician’s statement and notes written by the assessor during a home visit. There are seven levels of care: Support Levels 1 and 2 (i.e., individuals who require daily assistance) and Care Need Levels 1 to 5 (i.e., individuals who are bedridden or in need of assistance for daily activities, such as eating and bathing). Once approved for LTC, they are eligible to take a monetary amount of services according to their level of functional disability and dementia.

Several studies have shown associations between recent LTC insurance certification as a proxy for functional disability and dementia and various risk factors, including physical and cognitive function, nutrition, laboratory findings, and social factors. However, a scientific review of risk factors for functional disability and dementia based on the LTC insurance system has not yet been conducted from the perspective of community-based studies. Therefore, we aimed to bridge this gap by systematically reviewing longitudinal studies concerning risk factors associated with LTC insurance certification in Japan.

## 2. Materials and Methods

### 2.1. Protocol and Registration

The review protocol was drafted according to the PRISMA Extension for Scoping Reviews (PRISMA-ScR) and reviewed by the authors. The final protocol was registered with the Open Science Framework and is available on the project page (https://osf.io/5jyem/ (accessed on 29 June 2021)). The PRISMA-ScR checklist is provided in Appendix B.

### 2.2. Study Inclusion Criteria

Articles for this review were included if they met the following eligibility criteria: published between April 2000, when the Japanese LTC system started, and December 2019; written in English or Japanese; followed a prospective/retrospective cohort or nested case–control design to identify risk factors and estimate causality; the endpoint was LTC certification; participants were recruited from community settings.

### 2.3. Literature Search

The PubMed, CINAHL, and EMBASE databases were searched for studies published between April 2000 and December 2019. The final search strategy for PubMed is available in Appendix C.

### 2.4. Data Selection

Two authors (ST and KT) independently read the titles and abstracts of the identified studies to determine whether they should be included in this review. Studies were included when both authors agreed that these met the inclusion criteria. In cases of disagreement, a decision was made after a discussion between both authors (Figure 1).

We identified 647 studies: 628 from the database search and 19 from the manual search. Of these, 299 studies were excluded because of duplication. After reading the titles and abstracts, 263 papers were excluded because they did not follow a cohort or nested case–control design (*n* = 210), the endpoint was not LTC certification (*n* = 23), study participants were not recruited from the community-dwelling population (*n* = 20), or they were duplicates (*n* = 10). Therefore, 85 papers were eligible for full-text review. Of these, one paper [11] was excluded from the analysis because its outcome was the trajectory of the Care Need Level. We examined the reference lists of the included studies to identify additional literature, and another seven articles were added. Ultimately, 91 studies were included in the analysis.

### 2.5. Endpoints and Risk Factors

Previously, LTC certification has been used as a proxy for the incidence of functional disability and/or dementia in studies. For this review, the endpoints of the reviewed studies were classified into two categories: functional disability and dementia. We investigated the levels of support and care provided by the LTC insurance system, the criteria for determining the onset of dementia, and the combined endpoint of LTC and death. Risk factors were categorized as physical condition, lifestyle, dental status, medical history, blood tests and clinical examinations, social factors, Kihon Checklist scores (a predictive tool for disability), and other.

## 3. Results

Of the 91 studies included in the analysis, 84 were prospective cohort studies, 1 was a retrospective cohort study, and 6 were nested case–control studies (Appendix A). Seventy-two studies used LTC certification as an index of functional disability (Table 1). Levels of certification for LTC were highly prevalent: 64 studies focused on LTC Support Level ≥ 1, 7 on LTC Care Need Level ≥ 2, and 3 on Care Need Level ≥ 3. Overall, 14 reports used LTC certification or death as the endpoint, and 22 used LTC certification as a proxy for the onset of dementia. Criteria for the onset of dementia were as follows: 20 studies referred to a degree of independence of II or higher among older adults with dementia, as determined by the accreditation, and 2 studies based disease classification on the attending physician’s opinion.

Among the studies that considered LTC certification as a proxy for functional disability, the most common risk factor was physical condition (e.g., motor function, physical frailty, and sarcopenia; 15 studies), followed by medical history (14 studies), blood tests and clinical examinations (14 studies), and social factors (14 studies; Table 2). Among studies using LTC as a proxy for the onset of dementia, lifestyle factors such as diet and smoking were the most common risk factors (seven studies).

### 3.1. Physical Function

Motor function was mainly measured objectively and included assessments of grip strength [12,13], knee extension torque [12], usual gait speed (6-m walk) [12], chair stand time [12], muscle dysfunction [12,14], the timed up and go (TUG) test [13], and one-leg standing time with eyes open [14].

Makino et al. showed that step count and moderate- to vigorous-intensity physical activity were associated with an increased disability risk [15]. Akune et al. reported that physical activities of daily living (ADLs) as assessed by the Western Ontario and McMaster Universities Osteoarthritis Index (WOMAC), a self-administered questionnaire used to measure patients’ subjective function and pain status, were a predictor of functional disability [16]. Fujiwara et al. found that poor walking ability was associated with functional disability [17] and poor instrumental ADLs were associated with severe functional disability (Care Need Level ≥ 2) [17]. Hirai et al. showed that daily walking time and frequency of going out were significantly associated with a higher incidence of functional disability [18]. Tsutsumimoto et al. demonstrated that slow gait speed and/or depression (assessed by the Geriatric Depression Scale (GDS)) were significantly associated with greater functional disability risk [19]. Hoshi et al. showed that the predictive power of physical function, assessed using the Motor Fitness Scale questionnaire, was equivalent to that assessed using physical performance measures, such as maximum walking velocity, the TUG test, leg extension power, and the functional reach test [20].

Chen et al. suggested that prefrail and frail individuals, measured by two frailty criteria from the Cardiovascular Health Study and a simple frailty questionnaire (FRAIL), had a significantly higher risk of functional disability [21]. Makizako et al. [22] and Shimada et al. [23] focused on the various components of frailty. Some components, such as slowness, weakness, and weight loss, were strongly associated with future disability [22], while prefrailty with slow walking speed and frailty with and without slow walking speed were associated with a higher risk of disability [23].

Regarding sarcopenia, Tanaka examined the validity of the “Yubi-wakka” (finger-ring) test for the swift assessment of sarcopenia as a practical method to identify severe functional disability (Care Need Level ≥ 3) [24]. Participants whose calf circumference was smaller than their finger-ring circumference had increased disability risk [24].

Fujiwara reported that poor chewing ability was predictive of functional disability only in women [17]. Hirai et al. showed a substantial association between biting ability and functional disability, regardless of sex [18]. Moriya et al. demonstrated that participants aged 65–79 years with fair or poor masticatory ability had a significantly higher risk of functional disability than those with good masticatory ability [25].

Based on a questionnaire and incidence of functional disability, Matsunaga et al. found no significant association between the amount of leisure-time physical activity and functional disability but did show an inverse dose–response relationship between the amount of leisure-time physical activity and incidence of dementia in men [26].

Tomata et al. examined the association between changes in time spent walking and incidence of disabling dementia [27]. People who spent at least 1 h walking per day had a lower risk of dementia than those who spent less than 30 min walking per day. Additionally, the population attributable fraction (PAF) for dementia was 18.1% if all participants spent at least 1 h walking per day and 14.0% if participants increased their daily walking time by one level [28].

Hirai et al. indicated that falling in the past year was significantly associated with functional disability in both sexes [18]. Makino et al. showed that those with a fear of falling, either with or without a fall history, had a higher incidence of functional disability than those with neither a fear of falling nor a fall history [29].

Hirai et al. demonstrated an association between excretion impairment and a higher risk of functional disability in both sexes [18].

### 3.2. Lifestyle Factors

Ikeda et al. revealed an association between smoking and dementia [30], where current smokers were at twice the risk of dementia compared with those who had never smoked. Noguchi-Shinobara et al. clarified that current smokers with diabetes were at a greater risk of dementia than never smokers without diabetes [31].

In the Ohsaki Cohort Study, green tea consumption was associated with a lower risk of functional disability [32]. Furthermore, in an analysis combining the data sets from the Ohsaki Cohort Study and Ohsaki Cohort 2006, Matsuyama et al. and Tomata et al. showed that improved adherence to the Japanese diet was associated with a lower risk of functional disability, regardless of initial adherence status [33,34]. Japanese dietary patterns, citrus consumption, mushroom consumption, and coffee consumption were all shown to be associated with a lower risk of dementia [35,36,37,38,39].

Lu et al. showed that individuals with long sleep duration had a higher incidence of dementia [40]. Zhang et al. revealed that shorter sleep duration (<6 h/day) was associated with a higher risk of functional disability when considering death as a competing event [41].

Zhang et al. examined the effect of a combination of healthy lifestyle behaviors, including smoking status, time spent walking, and consumption of fruit and vegetables [42,43]. Their findings showed that adherence to each additional healthy lifestyle behavior resulted in a PAF of 10.5% for disability reduction [42]. They also showed that those who adhered to all three healthy lifestyle behaviors lived an average of 17.1 months longer without functional disability [43].

Yagi et al. showed that a high frequency of bathing (≥7 times/week) was associated with a lower risk of functional disability compared to a low frequency of bathing (0 to 2 times/week) [44].

### 3.3. Dental Condition

Aida et al. showed that respondents with ≤19 teeth had a higher risk of functional disability than those with ≥20 teeth [45]. Komiyama et al. reported that participants without regular dental appointments had a significantly higher incidence of functional disability than those with regular dental appointments [46]. They also showed that participants with 10–19 teeth, 1–9 teeth, or no teeth were more likely to develop functional disability than those with ≥20 teeth [47]. They assessed dentition status using Miyachi’s Triangular Classification and showed that patients with 10 remaining teeth and patients with 4 occlusal supports and 11 remaining teeth were more likely to develop functional disability 3 or more years after the baseline survey [48]. Ohi et al. found that a lower maximum occlusal force was significantly associated with an increased risk of functional disability [49]. Bando et al. revealed that frequent toothbrushing decreased the incidence of functional disability [50]. Yamamoto et al. demonstrated that the risk of dementia onset was significantly higher in those with few teeth and without dentures and those who did not have regular dental checkups, compared to others [51].

### 3.4. Medical History

Hirai et al. noted that currently being treated for a disease represented a risk factor for functional disability [18]. However, Fujiwara found that history of hospitalization during the past year was predictive of functional disability only in women [17].

Nitta et al. showed that participants with peripheral arterial disease, especially those with low physical function, had a higher risk of functional disability [52]. Jinnouchi et al. showed that retinal vascular changes, such as generalized arteriolar narrowing and two or more retinal abnormalities, were associated with an increased risk of dementia [53].

Yamada et al. demonstrated that participants with an estimated glomerular filtration rate (eGFR) < 60.0 mL/min/1.73 m^2^ had an elevated risk of functional disability [54]. Lee et al. showed that lower eGFR was a risk factor for functional disability among those who had ≥8 h of sedentary time per day [55]. Himeno et al. showed that chronic kidney disease (CKD) patients with moderate systolic blood pressure (130–159 mmHg) were at a lower risk of functional disability and dementia compared to those with low systolic blood pressure (<130 mmHg); however, these relationships were not found in non-CKD patients [56]. Watanabe et al. showed that CKD, diabetes, and lower HbA1c comorbidity was a strong predictor of functional disability [57].

Nishiguchi et al. demonstrated that cognitive decline, as evaluated by the Cognitive Performance Scale, was associated with an increased risk of functional disability [58]. Taniguchi et al. demonstrated a dose–response relationship between changes in Mini-Mental State Examination score per year and incidence of severe functional disability (Care Need Level ≥ 2) [59]. Fujiwara et al. showed that severe cognitive decline was a predictor of severe functional disability (Care Need Level ≥ 2) only in men [17]. Makizako et al. showed that cognitive function and depressive symptoms were associated with functional disability incidence [22]. Shimada et al. showed that psychological frailty, defined as the copresence of physical frailty and depressive mood (GDS-15), was associated with a higher risk of functional disability [60].

Four papers focused on psychological factors associated with functional disability. Tomata et al. [61] demonstrated a dose–response relationship between psychological distress, measured by K6 scores, and incidence of functional disability. Yamazaki et al. and Ohmori-Matsuda et al. showed that a depressive state, as assessed by the GDS, was a risk factor for functional disability [62,63]. Hirai et al. revealed a sex difference in terms of severe functional disability (Care Need Level ≥ 2) and depressive state as assessed by the GDS [18].

Bae et al. showed that hearing problems and a lack of social activity were independent risk factors for developing functional disability [64]. Hirari et al. found that, while visual impairment was significantly related to the incidence of severe functional disability in men (Care Need Level ≥ 2), hearing problems were related to severe functional disability in women [18].

Makino et al. demonstrated that severe pain was associated with a higher risk of functional disability than mild pain [65].

### 3.5. Blood Tests and Clinical Examinations

The Tsurugaya Project reported relationships between blood test data and functional disability in five studies. The first study identified an inverse linear relationship between serum albumin levels and functional disability or death [66]. The second study on plasma NT-pro BNP levels showed that participants in the 90th percentile (241 pg/mL) or with higher plasma NT-pro BNP levels had a significantly higher risk of functional disability or death than those in the 25th percentile [67]. The third study showed that the relationship between adiponectin and the composite outcome of incident disability and death was partly explained by reduced physical function and wasting in participants with higher adiponectin levels (≥22.4 mg/L) [68]. The fourth study showed that higher equol levels (≥23.6 ng/mL), but not any other isoflavones, were inversely associated with an increased risk of functional disability or death [69]. The fifth study showed that lower levels of serum total cholesterol (<177 mg/dL) were significantly associated with an increased risk for functional disability, compared with 212–230 mg/dL [70].

The Circulatory Risk in Communities Study (CIRCS) conducted three studies on the association between blood markers and dementia. The first showed that serum coenzyme Q10 levels were inversely associated with the risk of dementia [71]. The second showed that elevated high-sensitivity C-reactive protein levels were associated with increased risk of dementia in individuals with a history of stroke, but not in those without [72]. The third showed that serum alpha-linolenic acid levels were inversely associated with the risk of dementia [73].

In another study on blood markers, Doi et al. found that lower serum IGF-1 levels were significantly associated with an increased risk of functional disability [74]. Further, Yamazaki et al. demonstrated that low ALT (<20 IU/L) was significantly associated with an increased risk of functional disability (Care Need Level ≥ 2) or death [75].

Takahashi et al. showed that urine albumin–creatinine ratio was a predictor for incidence of functional disability, even after adjusting for the onset of cardiovascular disease during the follow-up period [76]. Okuno et al. showed that subclinical major electrocardiographic abnormality predicted a higher risk of functional disability among older adults with no prior history of cardiovascular disease [77].

Although the cut-off values differed among studies, being underweight was identified as a risk factor for functional disability. Honda et al. showed that the young-old (65–74 years) with a body mass index (BMI) < 18.5 kg/m^2^ had a higher risk of functional disability compared to those with a BMI of 18.5 to <25.0 kg/m^2^ [78]. Zhang et al. showed that a BMI < 23 kg/m^2^ was a risk factor for functional disability due to dementia and a BMI ≥ 29 kg/m^2^ was a risk factor for functional disability due to joint disease [79]. Using 10-year follow-up data from the same cohort, they also showed that disability-free survival was 7.8 to 25.6 months shorter in those with a BMI < 23 or ≥29 kg/m^2^, compared with 25−27 kg/m^2^ [80].

### 3.6. Social Factors

Hirai et al. showed a significant association between participation in group activities and functional disability [18]. Ashida et al. demonstrated those who participated in a sports or hobby group, or who were group facilitators, were less likely to develop a disability, and the associations were stronger among highly educated older adults [81]. Otsuka et al. indicated that cognitive activity and time spent walking modified the association between social participation and incident functional disability [82].

Nemoto et al. showed that those who participated in social activities had a significantly lower risk of dementia than those who did not, among both the young-old and old-old [83]. In addition, this study demonstrated that those holding leadership positions had a significantly lower risk of dementia than regular members [83]. Saito et al. demonstrated that participating in community groups (e.g., sports groups) or engaging in paid work was significantly associated with decreased dementia risk [84].

Hirai et al. assessed receiving and providing emotional and instrumental support as social support, and frequency of contact with friends and going out as social characteristics [18]. Yokokawa et al. demonstrated that being homebound, operationally defined as walking outdoors for <5 min/day, was a risk factor for functional disability in women but not in men [85]. Makizako et al. demonstrated that social frailty, assessed using simple questions regarding living alone, frequency of going out compared with the previous year, visiting friends, feeling helpful to friends or family, and talking with someone every day, was associated with an increased risk of functional disability [86]. Saito et al. showed that social relationship diversity scores (range: 0 to 5) using social relationship domains (social support, social networks, and social activities) had an inverse linear relationship with dementia incidence [84]. Noguchi et al. showed that higher levels of community social cohesion reduced the risk of functional disability (Care Need Level ≥ 2) in men but not in women [87].

Saito et al. showed that older adults living only with their children had a significantly higher risk of functional disability than those living in three-generation households [88]. Saito et al. showed that men with nonspousal cohabitants and those living alone had a higher risk of functional disability than those living with a spouse, whereas there were no significant associations in women [89].

Momosaki et al. showed that low subjective food store availability was associated with a high risk for functional disability [90]. Tani et al. also reported that lower objective and subjective food store availability was associated with a high risk of dementia [91].

Hirai et al. showed that education ≤9 years was associated with an increased risk of functional disability in men, but not in women [18]. Takasugi et al. demonstrated that people of both sexes with <6 years of education had a higher risk of dementia than those with ≥13 years of education. In women, but not in men, there was an association between equivalized household income and dementia risk [92]. Nurrika et al. showed that a higher educational level (upper secondary education and above) was negatively associated with the incidence of functional disability. Participation in community activities contributed moderately to the relationship between education level and functional disability in those aged 65–74 years [93].

Kondo et al. indicated an association between relative deprivation, calculated using the Yitzhaki Index, and functional disability in men but not women. In addition, men with higher incomes may be more vulnerable to relative deprivation than those with lower incomes [94]. Kondo et al. demonstrated that the intensity and attitude domain of *Mujin*, Japanese rotating savings and credit associations, was associated with a low risk of functional disability (Care Need Level ≥ 3), whereas the financial domain of *Mujin* had negative effects on incident disability [95].

### 3.7. The Kihon Checklist as a Predictive Tool for Disability

The KCL consists of 25 items categorized into seven structured domains: “20 items other than five related to depressive mood domain”, “physical functions”, “nutritional state”, “oral function”, “homebound state”, “cognitive function”, and “depressive mood”. Tomata et al. demonstrated that all domains of the KCL were associated with the risk of 1-year incidence of functional disability [96]. Kamegaya et al. showed a higher risk of 3-year incidence of functional disability in four domains: “physical function”, “nutritional state”, “cognitive function”, and “depressive mood” [97]. Okabe et al. showed that the KCL predicted health life expectancy better (Care Need Level ≥ 2 or death) than the items related to specific health checkups, such as urine protein [98]. Satake et al. found that KCL classification of physical frailty status could be a significant tool to predict the 3-year risk of functional disability [99]. Katsura et al. showed that many, but not all, items were significantly associated with a 5-year risk of functional disability among both the young-old and old-old [100]. Shinkai et al. validated the checklist against the original version for screening high-risk older adults [101].

### 3.8. Other

Hirai et al. demonstrated that instrumental self-maintenance, effectance, and social role, as assessed by the Tokyo Metropolitan Institute of Gerontology Index of Competence (TMIG-IC), were associated with the incidence of functional disability in both sexes [18].

Kotaki et al. examined the joint impact of seven risk factors: diabetes mellitus, hypertension, obesity, physical inactivity, severe psychological distress, smoking, low educational attainment, and dementia. They found a dose–response relationship between the total number of risk factors and incident dementia and that participants had improved toward a better category, PAF 23.0% [102].

## 4. Discussion

Different endpoints of functional disability were assigned in each study, such as LTC Support Level ≥ 1 or LTC Care Need Level ≥ 2. Previous studies have shown that LTC certification level is closely related to ADL performance [103]. LTC Care Need Level 2 was defined as the requirement of assistance in at least one basic ADL task [50]. Some studies defined individuals with LTC Care Need Level ≥ 3 as having severe functional disability, while others used LTC Care Need Level ≥ 2 as an indicator of functional disability. Since each level of LTC insurance differs for physical disability and cognitive function, further studies are needed to definitively determine severe functional disability.

Twenty articles used a rank of II or higher on the dementia scale in primary doctors’ opinion papers for the onset of dementia. This cut-off point is consistent with the cut-off evaluated by the gold standard in Meguro et al. [5] and Noda et al. [104]. However, although information on LTC can be used as an index for dementia onset, one limitation might be that actual causes of dementia, such as cerebrovascular disease, could not be determined. 

Overall, motor function has been reported to be consistently associated with incident functional disability and dementia through objective and subjective measurements. However, the actual amount of physical activity and daily exercise level, along with changes in activity, have not yet been fully evaluated. Physical frailty, prefrailty, and social frailty, including KCL items, were well investigated in the analyzed articles. Combining physical and social frailty may bolster the effectiveness of preventive measures prior to LTC certification. 

In most studies, modifiable risk factors, such as drinking, smoking, obesity, being underweight, nutrition, and physical activity were analyzed as covariates. More detailed analyses of smoking and drinking behaviors might support growing evidence for LTC (e.g., the number of cigarettes smoked or smoking cessation). The presence or incidence of disease is a mediating factor between lifestyle factors and LTC. Adjusting for these diseases sequentially may help to identify major factors associated with LTC. Blood tests and clinical examinations may be used as screening tools to estimate future LTC needs. Although social factors have been examined, stratified analyses should be considered to identify vulnerable groups.

Some articles indicated sex differences in risk factors for future LTC. For example, the relationship between depressive states and functional disability was significant only in men in the study of Ohmori-Matsuda et al. [63]. Investigating sex differences between risk factors and LTC may provide insight into sex differences in longevity. Furthermore, risk groups could potentially be identified by combining several risk factors with conventional risk factors [42,43].

Most studies in this review were conducted among individuals aged 65 or older. Several studies indicate that lifestyles in midlife contribute to a decline in activities of daily living and cognitive function in later life [105,106,107]; however, few studies have summarized the association with LTC insurance certification. Thus, the association between lifestyle factors, blood tests, clinical examinations, and LTC among middle-aged people remains unclear. In terms of LTC prevention, life-course approaches are needed to clarify the midlife risks of LTC insurance certification to predict their later life. Consequently, interventions and cohort studies with long-term follow-up are needed among middle-aged people.

The main purpose of this study was to provide an overview of the status of longitudinal observational studies using endpoints related to LTC certification. Thus, studies that employed changes or trajectories of levels in LTC as outcomes were not included in this review. Since LTC assessment was correlated with functional disability and cognitive function in previous studies [94,103], an assessment of LTC allows us to evaluate patients’ general status of physical and cognitive disabilities on a scientific basis. In addition, all candidates who have a functional disability and/or dementia and seek to apply for services must be assessed by officials in the local government in Japan. Given that policymakers can verify the effect implemented, LTC is applicable for evaluating cost and effectiveness as a health care policy. Global policymakers should consider these risk factors when seeking to implement effective preventive measures against functional disability and intervene sooner.

LTC insurances are provided by the Government of Japan to disabled elderly, including those with physical disabilities and those with cognitive dysfunctions. Meanwhile, several care models are applied to the elderly worldwide. For example, the chronic care model is widely used to improve chronic patient outcomes by changing patients’ care and managing chronic diseases [6]. The chronic care model is compatible with LTC insurance in terms of availability to take care and services, particularly for functional disability, as one of the major endpoints of LTC in the present review. Therefore, the results in the endpoints of LTC could be interpreted in the international context by partially predicting those who need support in the chronic care model.

This review has some limitations. First, the variation in methods for exposure measurement was not examined. Furthermore, we did not verify the validity of the instruments or assess the quality of the included studies. Second, the magnitude of factors associated with LTC could not be elucidated. Accumulating studies through meta-analyses might help to understand the differences in magnitude among the risk factors. In Japan, to prevent frailty and sarcopenia, the Ministry of Health, Labour and Welfare promoted the integration of LTC prevention services with health care strategies in April 2020 [108]. The project recommends using a 15-item questionnaire (including general health perception; life satisfaction; and physical, cognitive, mental, social, and oral frailty) to evaluate health conditions and identify frail persons among older people. Despite these limitations, to the best of our knowledge, there has been no scientific review of risk factors for the LTC insurance system in Japan. Furthermore, we showed two different endpoints regarding functional disability and dementia, which might provide important findings regarding the effectiveness of implementing modifiable risk factors for either functional disability or dementia. Therefore, we believe that this review concerning risk factors of LTC insurance certification has helped to clarify the scientific evidence for promoting the project.

## 5. Conclusions

We conducted a scoping review of the association between various risk factors and LTC insurance certification as a proxy for functional disability and dementia. Although risk factors such as motor function and frailty were assessed in most reports, these mechanisms were not fully investigated. Further studies should work toward developing preventive measures for functional disability.

## Figures and Tables

**Figure 1 ijerph-19-02162-f001:**
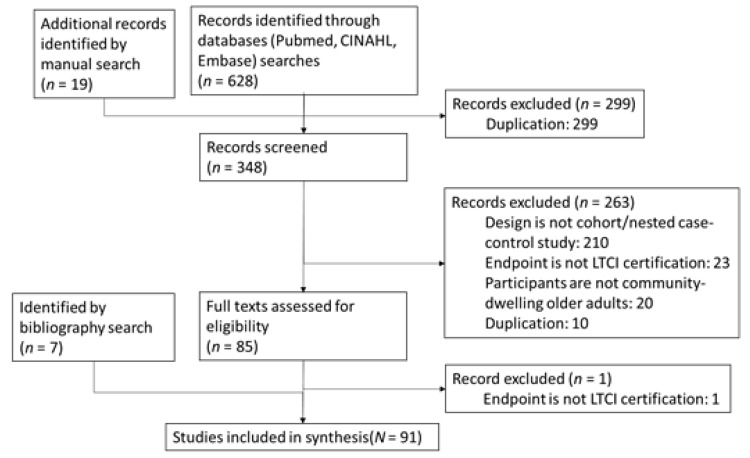
Flowchart of the literature review.

**Table 1 ijerph-19-02162-t001:** Classification of endpoints in the reviewed articles.

	*n*	%
Functional disability	72	
All Support/Care Need Levels	64	88.9%
Above specific level	10	13.9%
Care Need Level 2	7	9.7%
Care Need Level 3	3	4.2%
Composite endpoint with death	12	16.7%
Analysis on cause of disability	2	2.8%
Stroke	2	2.8%
Joint	2	2.8%
Other	1	1.4%
Dementia	22	
Dementia scale rank II or above	20	90.9%
Primary physician’s comment	2	9.1%
Composite endpoint with death	2	9.1%
Analysis of cause of disability	4	18.2%
Stroke	4	18.2%
Joint	0	0.0%
Other	0	0.0%

Studies that employed both functional disability and dementia as outcomes were counted in each category.

**Table 2 ijerph-19-02162-t002:** Number of articles by combination of endpoint and risk factor.

Risk Factor		Endpoint		
Category		Functional Disability	Dementia	Total
Physical condition		15	3	17
	Motor function	9	0	9
	Physical frailty and sarcopenia	4	0	4
	Chewing ability	3	0	3
	Leisure-time physical activity	1	1	1
	Walking	0	2	2
	Falls	2	0	2
	Other physical function	1	0	1
Lifestyle		6	7	13
	Smoking	0	2	2
	Nutrition	3	5	8
	Sleep	1	1	2
	Combination of healthy lifestyle behaviors	2	0	2
	Other lifestyle factors	1	0	1
Dental status		7	1	8
Medical history		17	2	18
	Hospitalization	2	0	2
	Vascular diseases	1	1	2
	Chronic kidney disease	4	1	4
	Cognitive dysfunction	5	0	5
	Psychological difficulties	4	0	4
	Sensory organ abnormalities	1	0	1
	Pain	1	0	1
Blood tests and clinical examinations		12	4	15
	Blood tests	7	3	10
	Clinical examinations (except blood tests)	2	0	2
	Body mass index	3	1	3
Social factors		13	3	16
	Social participation, social support, social capital, and social frailty	4	1	5
	Social network	4	0	4
	Living arrangements	2	0	2
	Food availability	1	1	2
	Other social factors	4	1	5
Kihon Checklist as a predictive tool for disability		6	0	6
Others		1	1	2

Studies that corresponded to multiple categories were counted in each category. Thus, the sum of categories does not match the total number of reviewed cases (N = 91). Even if one study corresponded to multiple subcategories within the same category, the category count was counted as one case. Therefore, the sum of the subcategories does not match the frequency of the category.

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
