# Peer review of "Risk Factors of Long-Term Care Insurance Certification in Japan: A Scoping Review"

_ijerph, 2022, doi:10.3390/ijerph19042162_

Round 1

Reviewer 1 Report

Although the paper is well designed and correctly written, I consider its: (1) novelty level; (2) interests to readers and (3) implication arising from it, as  too simplistic for IJERPH. Therefore, I do not recommend to publish this article at IJERPH. I recommend, the authors to look for another journal, with significantly lower impact factor.

Reviewer 2 Report

The manuscript presents a meticulous review of risk factors associated with long term care certification in Japan. As far as I am concerned, the article needs no particular major changes. However, it might generate more interest if the authors expand the introduction and discussion by going beyond the specific topic of LTC insurance.For example, the introduction could include references to the internationally known Chronic care model. It could also be interesting to better specify, in the purpose of the research, why it is relevant on a scientific or applicative level to validate LTC insurance as a proxy endpoint.These points could be taken up in the discussion, to allow the reader to extend the interest of the research beyond the Japanese context

Reviewer 3 Report

Thanks for the opportunity to review this important piece of work on impact of long term care insurance certification in elderly population in Japan. Just a few questions from me:

1) I am assuming the search start date of April 2020 ties in with the launch of long term insurance system implemented in Japan. Just clarifying that point.

2) The end of search date is in December 2019 and it's been more than 2 years. I am wondering whether the authors have considered to update the literature search to include more recent publications.

3) I can understand the LTC system is implemented in Japan, perhaps unique to the local context. Wondering how relevant the findings to the international context as we have a rapidly aging population.

4) Physical disability affecting motor function, frailty and sarcopenia not surprisingly top the chart, having the most influence in increasing LTC certification. In Japan, are there any existing national programs to tackle frailty and sarcopenia which are conditions catching the limelight recently? Lifestyle factors may be the ones who are modifiable, any thoughts about interventions which can be effective in slowing down frailty and increase intrinsic capacity in these elderly group. That may have a significant impact in reducing LTC certification.

Round 2

Reviewer 3 Report

Happy with the answers provided for my questions. Cheers.